



# Do phenomenological dynamical paleoclimate models have physical similarity with Nature? Seemingly, not all of them do.

Mikhail Y. Verbitsky[1,2] and Michel Crucifix[2]

[1]Gen5 Group, LLC, Newton, MA, USA
[2]UCLouvain, Earth and Life Institute, Louvain-la-Neuve, Belgium

Correspondence: Mikhaïl Verbitsky (verbitskys@gmail.com)

**Abstract.** Phenomenological models may be impressive in reproducing empirical time series but this is not sufficient to claim physical similarity with Nature until comparison of similarity parameters is performed. We illustrated such a process of diagnostics of physical similarity by comparing the phenomenological dynamical paleoclimate model of Ganopolski (2023), the van der Pol model (as used by Crucifix, 2013), and the model of Leloup and Paillard (2022) with the physically explicit Verbitsky *et al* (2018) model that played a role of a reference dynamical system. We concluded that phenomenological models of Ganopolski (2023) and of Leloup and Paillard (2022) may be considered to be physically similar with the proxy parent dynamical system in some range of parameters, or in other words they may be derived from basic laws of physics under some reasonable physical assumptions. We have not been able to arrive to the same conclusion regarding the van der Pol model. Though developments of better proxies of the parent dynamical system should be encouraged, we nevertheless believe that the diagnostics of physical similarity, as we describe it here, should become a standard procedure to delineate a model that is merely a statistical description of the data, from a model that can be claimed to have a link with known physical assumptions.

## 1. Introduction.

A mathematical model that is constructed to understand a physical phenomenon must be simple enough, otherwise the interpretation of the modeling results may be as difficult as the interpretation of direct observations. In that regard, even most sophisticated space-resolving models of global climate provide, indeed, a simplified picture of the phenomenon, but a much more drastic degree of simplification is required when we study climate on timescales of tens of thousands of years. Faced with this challenge, Barry Saltzman used to be a proponent of the phenomenological approach "through the construction of low-order models in which the full behavior is projected onto the dynamics of a reduced number of …highly aggregated variables…" (Saltzman, 2002). Phenomenological models of paleoclimate variability have routinely been used to explain certain characteristics of glacial-interglacial cycles (e.g., Saltzman and Maasch, 1991, Saltzman and Verbitsky, 1992, 1993, 1994, Paillard, 1998, Tziperman et al, 2006, Crucifix, 2013, Kaufmann and Pretis, 2021, Talento and Ganopolski, 2021, Leloup and Paillard, 2022, Ganopolski, 2023). The core principle of the phenomenological approach is to fit model-produced time series to the observational time series. When this goal is achieved, it is tacitly assumed that there must be some physical similarity between the phenomenological model and Nature. We believe though that the assumption of physical similarity with Nature can be more rigorously challenged before the implications of a phenomenological model are accepted.

In fluid dynamics, for example, the concept of physical similarity is the cornerstone of any judgement built on model experimentations. Classical similarity parameters, which emerge from the analysis of fundamental conservation laws, like the Reynolds number, the Peclet number, the Euler number, etc., quantify the relative importance of different aspects of fluid flow. For an experimental or a numerical model to be relevant, it should have quantitatively the same similarity parameters as those of the natural phenomenon being considered. We will now apply this concept of physical similarity to dynamical paleoclimate systems.

As physicists, we might want to describe a phenomenon such as ice ages as "emerging from fundamental laws". However, the fundamental laws that we know in physics dictate interactions between





particles. Perhaps one of the greatest challenges of the physical approach to complex systems is to explain
how Nature organizes billions of billions of particles in interaction to generate some predictable behavior
even on very long time scales such as, precisely, glacial-interglacial cycles. The methods of statistical
physics tell us how to define macroscopic variables to describe the collective behavior of particles
submitted to a conservation constraint, and how the phenomenon of dissipation emerges as a consequence
of statistical mixing in a chaotic system. Dynamical system's theory tell us why we mainly see the most
unstable modes of a system (Haken, 2006) and how time scale separation assumptions allows us to focus
on a subset of the system's variables. In a nutshell, the theories of mathematical and statistical physics
make it legitimate to assume that there is a natural parent dynamical system with much fewer degrees of
freedom than Avogadro's number, and which has generated the phemenon that we see.

What "much fewer" means is not a straightforward matter. It depends on what we describe as the
phenomenon, and how fine-grained this description is. For example, the successions of glacial-interglacial
cycles and the timing of deglaciations appear to follow fairly simple, predictable rules (Tzedakis et al.,
2017). Hence, it is legitimate to assume that the physical parent dynamical system, which dictates the
evolution of the macroscopic state of climate at the orbital time scale, can be reduced to a small number
of degrees of freedom.

Specifically, we may suggest that this parent dynamical system is governed by $n$ physical parameters
$a_i$ such that a dependent variable of interest, $x$, can be expressed as function

$$x = \varphi(a_1, a_2, \ldots, a_i, \ldots, a_n) \qquad (1)$$

If $k$ parameters of $a_1, a_2, \ldots, a_i, \ldots, a_n$ are parameters with independent dimensions, then, according to $\pi$-theorem (Buckingham, 1914), in the dimensionless form, the phenomenon (1) can be described by
$m = n - k$ adimensional similarity parameters $\Pi_1, \Pi_2, \ldots, \Pi_i, \ldots, \Pi_m$:

$$\Pi = \Phi(\Pi_1, \Pi_2, \ldots, \Pi_i, \ldots, \Pi_m) \qquad (2)$$

Two physical phenomena have physical similarity if both of them are described in the adimensional
form by the same function $\Phi(\Pi_1, \Pi_2, \ldots, \Pi_i, \ldots, \Pi_m)$ and have identical numerical values of similarity
parameters $\Pi_1, \Pi_2, \ldots, \Pi_i, \ldots, \Pi_m$, though numerical values of the governing parameters
$a_1, a_2, \ldots, a_i, \ldots, a_n$ may be different (e.g., Barenblatt, 2003).

As we have already mentioned, our knowledge about a parent dynamical system is suggested to us by
the presence of empirical time series. It means that one of the similarity parameters, let say $\Pi_1$, is
adimensional time $\frac{t}{\tau}$ ($t$ and $\tau$ are dimensional time and a timescale, correspondingly), and all other
parameters $\Pi_2, \ldots, \Pi_i, \ldots, \Pi_m$ are fixed to specific values. Hence, an experimental time series (neglecting
measure errors) can be described as

$$\Pi = \Phi\left(\frac{t}{\tau}, \Pi_2, \ldots, \Pi_i, \ldots, \Pi_m\right) \qquad (3)$$

If we created a model dynamical system such that it is governed by $p$ governing parameters $b_i$

$$x = \psi(b_1, b_2, \ldots, b_i, \ldots, b_p) \qquad (4)$$

and $r$ parameters of $b_1, b_2, \ldots, b_i, \ldots, b_p$ are parameters with independent dimensions, then, again,
according to $\pi$-theorem, in the dimensionless form, the model can be described by $q = p - r$
adimensional similarity parameters $\pi_1, \pi_2, \ldots, \pi_i, \ldots, \pi_q$:

$$\pi = \Psi(\pi_1, \pi_2, \ldots, \pi_i, \ldots, \pi_q) \qquad (5)$$

For a specific time series, and for a fixed set of parameters $\pi_2, \ldots, \pi_i, \ldots, \pi_q$, the model (5) can be
presented as



$$\pi = \Psi\left(\frac{t}{\tau}, \pi_2, \ldots, \pi_i, \ldots, \pi_q\right) \tag{6}$$
The essence of the phenomenological approach is to fit the function $\Psi\left(\frac{t}{\tau}, \pi_2, \ldots, \pi_i, \ldots, \pi_q\right)$ to the
function $\Phi\left(\frac{t}{\tau}, \Pi_2, \ldots, \Pi_i, \ldots, \Pi_m\right)$ under the "best" set of parameters $\pi_2, \ldots, \pi_i, \ldots, \pi_q$, i.e. to equate the
model time series $\Psi\left(\frac{t}{\tau}, \pi_2, \ldots, \pi_i, \ldots, \pi_q\right)$ and the natural, empirical, time series $\Phi\left(\frac{t}{\tau}, \Pi_2, \ldots, \Pi_i, \ldots, \Pi_m\right)$:
$$\Psi\left(\frac{t}{\tau}, \pi_2, \ldots, \pi_i, \ldots, \pi_q\right) = \Phi\left(\frac{t}{\tau}, \Pi_2, \ldots, \Pi_i, \ldots, \Pi_m\right) \tag{7}$$
It is obvious that even if the goal (7) is achieved at every $\frac{t}{\tau}$ - point, we still cannot claim the model (6)
to be physically similar to "Nature" (3) until we prove that $\pi_i = \Pi_i$, i.e., $\pi_i$-physics in the model is as
significant as the $\Pi_i$-physics of Nature. Simply speaking, merely *matching a proposed phenomenological*
*model with empirical data does not make a case for physical similarity* because it does not provide an
evidence that it happens for the right reason, the reason being the similarity parameters of the right value,
i.e., $\pi_i = \Pi_i$.
But how can we compare $\pi_i$-physics of the phenomenological model and $\Pi_i$-physics of Nature if the
phenomenological models are not derived from the laws of physics? Though, indeed, phenomenological
models have not been derived from the laws of physics, they are not completely ignorant of the physical
content: they still have a physical, measurable variable, time; they also have orbital and terrestrial
forcings as well as positive and negative feedbacks. If the parent dynamical system was formulated in
terms of similarity parameters formed by the ratios of timescales and by the ratios of the forcings' and
feedbacks' amplitudes, then the comparison with phenomenological models that also use time scales and
forcing and feedback ratios would be possible. The VCV18 model (Verbitsky et al, 2018), is one such
candidate (a proxy) for a parent dynamical system. VCV18 was *derived* from the scaled mass- and heat-
balance equations of the non-Newtonian ice flow. Next, we will derive scaling laws and similarity
parameters for three phenomenological models: (a) the model of Ganopolski (2023); (b) van der Pol
model as it has been described by Crucifix (2013); and (c) the model of Leloup and Paillard (2022);
G23,VDP and LP22, thereafter, respectively. Each of these models produces a specific function
$\Psi(\pi_1, \pi_2, \ldots, \pi_i, \ldots, \pi_q)$. We then compare functions $\Psi(\pi_1, \pi_2, \ldots, \pi_i, \ldots, \pi_q)$ of these models with the
corresponding function $\Phi(\Pi_1, \Pi_2, \ldots, \Pi_i, \ldots, \Pi_m)$ provided by VCV18 to recognize or reject the hypothesis
of physical similarity with a proxy for the parent dynamical system.
Certainly, we cannot expect that the time series produced by G23, VDP, and LP22 models and by the
VCV18 model are identical, and therefore these models will not be physically similar in the most rigorous
sense of the equation (7). We will demonstrate though that the answer to the physical-similarity question
is insightful if our dependent variable of interest $x$ is not necessarily a time series but a time-independent
attribute such as the period of glacial rhythmicity. All models of this study reproduce equally well ~100-
kyr period of the late Pleistocene glaciations. We will now evaluate if the similarity parameters involved
in the corresponding equations (7) are quantitatively the same.
**2.  Method**
**2.1 VCV18 model as a proxy for a parent dynamical system.**
Deriving a low-order dynamical paleoclimate model that may be considered as a candidate (a proxy)
parent dynamical system is not a trivial exercise. The "low-order" challenge means that out of the
multitude of physical phenomena involved only few should be recognized as dominant ones, and the
"dynamical" challenge means that the space-resolving properties should be sensibly reduced to some
integrated variables. Accordingly, in developing VCV18 proxy parent dynamical system, we first





assumed that ice ages can be explained by only two components of the global climate system, continental
ice sheets and the ocean representing the rest of the climate. For an ice sheet we adopted mass,
momentum, and heat conservation equations of a "thin" layer of homogeneous non-Newtonian ice, and
the rest of the climate was represented by the energy-balance equation. To migrate from the three-
dimensional to dynamical equations we used scaling analysis that provides simple mathematical
statements that are consistent with the original physics. Accordingly, the VCV18 dynamical model of the
ice-climate system is defined by the following set of equations:
$$\frac{dS}{dt} = \frac{4}{5}\zeta^{-1}S^{3/4}(\hat{a} - \varepsilon F - \kappa\omega - c\theta) \tag{8}$$
$$\frac{d\theta}{dt} = \zeta^{-1}S^{-1/4}(\hat{a} - \varepsilon F - \kappa\omega)\{\alpha\omega + \beta[S - S_0] - \theta\} \tag{9}$$
$$\frac{d\omega}{dt} = -\gamma[S - S_0] - \frac{\omega}{\tau} \tag{10}$$

Here, $S$ (m$^2$) is the area of glaciation, $\theta$ ($^{o}$C) is the basal ice-sheet temperature, and $\omega$ ($^{o}$C) is the global
temperature of the rest of the climate. Equation (8) represents global ice balance $\frac{d(HS)}{dt} = AS$, where the
ice thickness $H$ is determined from the thin-layer approximation of ice flow, $H = \zeta S^{1/4}$, $\zeta$ is dimensional
profile factor (Verbitsky and Chalikov, 1986) and $A = \hat{a} - \varepsilon F - \kappa\omega - c\theta$ is the surface mass influx.
Equation (9) describes vertical ice temperature advection with a time scale $H/(\hat{a} - \varepsilon F - \kappa\omega)$, and
equation (10) is the global energy-balance equation. The parameter $\hat{a}$ (m s$^{-1}$) is the snow precipitation
rate; $F$ is normalized external forcing, i.e., mid-July insolation at 65$^{o}$N (Berger and Loutre, 1991) of the
amplitude $\varepsilon$ (m s$^{-1}$); $\kappa\omega$ represents fast positive feedback from the global climate on ice-sheet mass
balance; $c\theta$ is the ice discharge due to ice-sheet basal sliding incorporing (both delayed due to the
vertical temperature advection) positive feedback from the global temperature, $\alpha\omega$, and a negative
feedback of basal temperature reaction to the changes of ice geometry $\beta[S - S_0]$. Further, $-\gamma[S - S_0]$ is
external forcing for global temperature (e.g., albedo); $\kappa$ (m s$^{-1}$ $^{o}$C$^{-1}$), $c$ (m s$^{-1}$ $^{o}$C$^{-1}$), $\alpha$ (adimensional), $\beta$
($^{o}$C m$^{-2}$) and $\gamma$ ($^{o}$C m$^{2}$ s$^{-1}$) are sensitivity coefficients; $S_0$ (m$^2$) is a reference glaciation area; and $\tau$ (s) is the
global-temperature timescale.
Schematically, the dynamical system (8) – (9) is shown in Fig. 1(a). It can be seen that the dynamics
of the VCV18 system is defined by the amplitude and periodicity of the orbital forcing, $\varepsilon, T$, by the
terrestrial forcing $\hat{a}$, and by three feedback loops: the fast positive feedback, $-\kappa\omega$, and by two delayed,
positive and negative feedbacks, combined in the term $-c\theta$. The dimensional analysis of the VCV18
model has been performed previously (Verbitsky and Crucifix, 2020, 2021, Verbitsky, 2022a). It was
revealed that its large-scale periodicity is generally governed by two dimensionless parameters: the ratio
of the astronomical forcing amplitude $\varepsilon$ to the terrestrial ice-sheet mass influx, $\Pi_2 = \varepsilon/\hat{a}$ and the so-called
$V$-number, $\Pi_3 = V$ that is the ratio of amplitudes of time-dependent positive and negative feedbacks.
Specifically, the period $P$ of the VCV18 system response to the astronomical forcing of period $T$ is of the
form (hereafter called the "$P$-scaling law"):
$$\frac{P}{T} = \Phi\left(\frac{\varepsilon}{\hat{a}}, V\right) \tag{11}$$
For $T = 40$ kyr, $\frac{\varepsilon}{\hat{a}} = 1.4$, $V = 0.7$, $\Phi = 2$ (obliquity-period doubling). The corresponding time series and
positive-versus-negative feedback evolution are shown in Fig. 2(a, b).






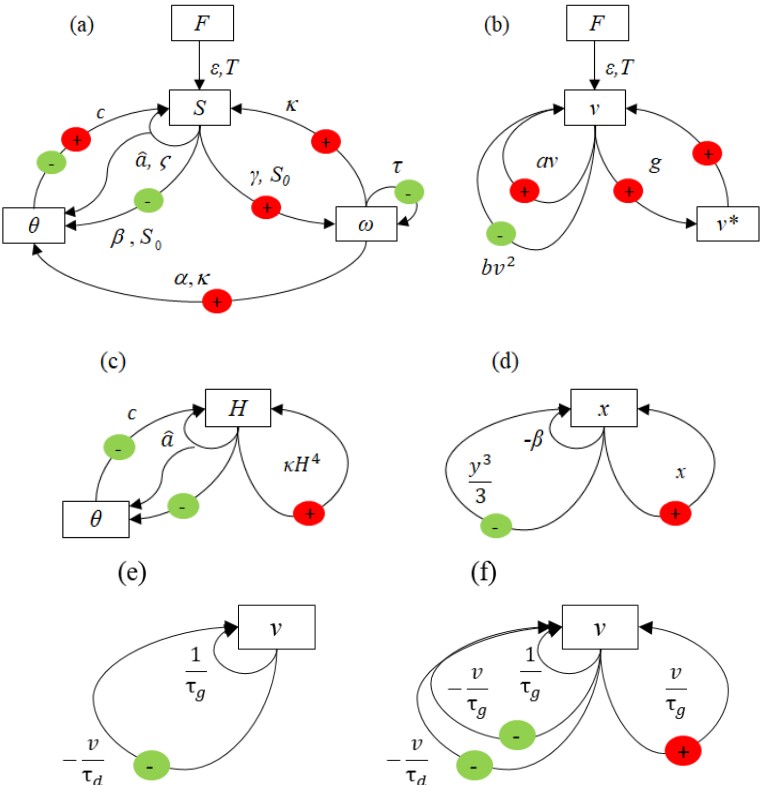


**Figure 1.** (a) The parent dynamical system VCV18 (Eqs. 8–10). Red circles mark positive feedback loops and green circles mark negative feedback loops; (b) The same for the G23 phenomenological model (Eqs. 12-13); (c) The same for simplified system VCV18-1 (Eqs. 22–23); (d) The same for VDP model (Eqs. 30-31); (e) The same for LP22 model (Eqs. 37-38), $I = 0$; (f) The same for LP22 model (Eqs. 44-45);

181

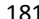

**Figure 2.** Time series (kyr BP) and corresponding positive-*vs*-negative feedback loops: (a, b) VCV18
(Eqs. 8–10), $\left(\frac{S}{S_0}\right)^{5/4}$ is normalized ice volume; (c, d) G23 (Eqs. 12-13); (e, f) VCV18-1 (Eqs. 22–23); all
variables are normalized by characteristic ice thickness, $H' = (\hat{a}/\kappa)^{1/4}$; the dotted triangle corresponds to
LP22 (Eqs. 37-38) without astronomical forcing; (g, h) VDP (Eqs. 30-31).





2.2 **G23 model**.
The G23 model describes evolution of global ice volume $v$ (adimensional) as a response to orbital forcing
$\varepsilon F$ ($F$ is normalized external forcing of the amplitude $\varepsilon$):
$$\frac{dv}{dt} = \frac{av - bv^2 - \varepsilon F + d}{1 - \delta g v^*}$$                                                      (12)

$$v^* = \frac{1}{T^*} \int_{t-T^*}^{t} v(t')dt'$$                                                       (13)

The term $\delta g v^*$ represents an additional positive feedback activated ($\delta = \delta_1 = 1$) when $\frac{dv}{dt} < 0$.
When $\frac{dv}{dt} \geq 0, \delta = \delta_2 = 0$. Graphically, the dynamical system (12) – (13) is shown in Fig. 1(b). We
observe that the dynamics of the G23 system is, like VCV18, defined by the amplitude and periodicity of
the orbital forcing, $\varepsilon, T$ and by three feedback loops: two positive feedbacks, $av, gv^*$ and by one negative
feedback, $-bv^2$. Unlike VCV18, though, all feedbacks are instantaneous. We now review how these
differences may be reflected in the corresponding $P$-scaling law.
For the purpose of dimensional analysis, we consider Eqs. (12) – (13) in dimensional form assuming
the following dimensions for variables and parameters involved: $t$ (s), $v$ (m³), $a$ (s⁻¹), $b$ (m⁻³s⁻¹), $\varepsilon$(m³s⁻¹), $F$
is an adimensional function of the period $T$ (s), $d$ (m³s⁻¹), $\delta_1, \delta_2$ (adimensional), $g$ (m⁻³), $v^*$(m³). The
period of the system response to the astronomical forcing is then a function of the following governing
parameters:
$$P = \psi(a, b, \varepsilon, T, d, \delta_1, \delta_2, g)$$                                                       (14)

For more explicit physical interpretation, instead of the parameter $b$, we will use parameter $\hat{a} = \frac{a^2}{b}$,
(m³s⁻¹), which is the mean growth rate. Also, for the reference values of parameters, provided by
G23, $d \ll \varepsilon$, and, lastly, $\delta_1, \delta_2$ are constant. Therefore we can re-write (14) as:
$$P = \psi(a, \hat{a}, \varepsilon, T, g)$$                                                       (15)

If we choose $\hat{a}, T$ as parameters with independent dimensions, then according to $\pi$-theorem:
$$\frac{P}{T} = \Psi\left(\frac{\varepsilon}{\hat{a}}, Ta, Tg\hat{a}\right)$$                                                       (16)

Let us now determine the $V$-number for G23 as the ratio of amplitudes of time-dependent positive and
negative feedbacks. Obviously, such ratio should be completely defined by the internal (terrestrial) G23
properties and therefore:
$$V = \lambda(a, b, g)$$                                                       (17)

If we choose $a$ and $b$ as parameters with independent dimensions, then according to $\pi$-theorem:
$$V = \Lambda\left(\frac{ga}{b}\right)$$                                                       (18)

We can also express $g$ as a function of $V$:
$$\frac{ga}{b} = \Lambda^{-1}(V)$$                                                       (19)



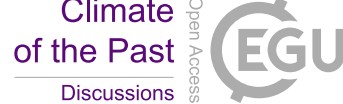

Accordingly, the *P*-scaling law of G23 can be written as:

$$\frac{P}{T} = \Psi \left[ \frac{\varepsilon}{\hat{a}}, Ta, Ta\Lambda^{-1}(V) \right]$$ (20)
or

$$\frac{P}{T} = \Psi \left( \frac{\varepsilon}{\hat{a}}, \frac{T}{\tau}, V \right); \ \tau = 1/a$$ (21)
We see that the scaling law (21) is different from the scaling law (11) because the $\Psi$ − function of (21)
depends on $T$ unlike the $\Phi$ − function of (11). There are only two scenarios for orbital periods to escape
the $\Psi$ − function (or the $\Phi$ − function) in a scaling law. First, they may be excluded from the governing
equations when the main period of system's variability is attributed to the internal, terrestrial, physics.
This is the case for VDP and LP22 models that will be considered later, but, definitely, it is not applicable
neither to G23 nor VCV18. The second scenario occurs when a system incorporates multiple parameters
encoding different time scales. The interplay of these parameters may create a situation when $T$-
dependent similarity parameters form jointly a $T$-independent conglomerate similarity parameter, giving a
system the so-called property of incomplete similarity (Barenblatt, 2003). This property has been
discovered for VCV18 (Verbitsky, 2022a). Indeed, it has two major timescales of the same order of
magnitude, the timescale of ice growth and the timescale of the vertical temperature advection in the ice
sheet. As the result, the $\Phi$ − function of the scaling law (11) does not depend on orbital period ($\Phi = 2$ in
the range of $T = 35 – 60$ kyr). Contrarily, as we have already noted, G23 positive and negative feedbacks
are instantaneous, G23 single ice-growth timescale $\tau \sim 1/a$ does not have a "counterpart" for an interplay,
and therefore the $\Psi$ − function of (21) is period-$T$ dependent.
Specifically, for $T = 40$ kyr, $\frac{\varepsilon}{\hat{a}} = 1.6$, $V = 1.1$, $\Psi = 2$ (obliquity-period doubling). Hence, only for a
given quasi-periodic forcing, e.g., obliquity, VCV18 and G23 models appear physically similar in regards
of two similarity parameters, $\frac{\varepsilon}{\hat{a}}$, the ratio of the astronomical forcing amplitude $\varepsilon$ to the growth rate $\hat{a}$,
and in terms of the *V*-number. The corresponding time series and positive-versus-negative feedback
evolution are shown in Fig. 2 (c, d).

**2.3 Simplified VCV18 model (VCV18-1) as a proxy for a parent dynamical system**

The next phase of our study is devoted to two phenomenological models, VDP and LP22 that have
100-kyr auto-oscillations independently of orbital forcing. To make the diagnostics of physical similarity
possible, we have to further simplify VCV18 system with several, physically reasonable, assumptions:
(a) Since the global-temperature timescale in equation (10) is much faster than other timescales
(orbital, ice accumulation, and ice-temperature advection), we assume that global temperature is
an instantaneous function of the glaciation forcing,
(b) In equation (9), we assume $\alpha = 0$ (for example, effect of increased global temperature is offset
by increased snow precipitation rate, see experiment D in the Appendix of VCV18), which
cancels the direct effect of climate on basal temperature,
(c) We rewrite equations (8) and (9) in terms of ice thickness $H = \zeta S^{1/4}$, and finally
(d) We attribute all system variability to terrestrial causes ($\varepsilon = 0$).
The simplified dynamical system then takes the following form:
$$\frac{dH}{dt} = \hat{a} + \kappa H^4 - c\theta$$ (22)



$$\frac{d\theta}{dt} = \frac{H^4 - \theta}{H/\hat{a}}$$ (23)

The physical meaning of all variables and governing parameters are the same as in equations (8) – (9), but
the numerical values and dimensions of some parameters and variables are, indeed, different. Specifically,
$t$(s), $H$ (m), $\theta$ (m$^4$), $\hat{a}$(m s$^{-1}$), $\kappa$(m$^{-3}$s$^{-1}$), $c$(m$^{-3}$s$^{-1}$). The casual graph of the dynamical system (22) – (23) is
shown in Fig. 1 (c).
The period of system variability is a function of three governing parameters:

$$P = \varphi(\hat{a}, \kappa, c)$$ (24)

If we choose $\hat{a}, \kappa$ as parameters with independent dimensions, then according to π-theorem:

$$\frac{P}{\tau} = \Phi\left(\frac{\kappa}{c}\right)$$ (25)

$$\tau = (\hat{a}^3 \kappa)^{-1/4}$$

Parameters $\hat{a}, \kappa, c$ lack in VDP and LP22 models, and therefore we transition to the *V*-number that must
be a function of the same $\hat{a}, \kappa, c$ parameters:

$$V = \lambda(\hat{a}, \kappa, c)$$ (26)

Since the *V*-number is adimensional, and $\hat{a}, \kappa$ are parameters with independent dimensions, then
according to π-theorem:

$$V = \Lambda(\kappa/c)$$ (27)

It also means that

$$\frac{\kappa}{c} = \Lambda^{-1}(V)$$ (28)

and we can finally present the VCV18-1 *P*-scaling law as:

$$\frac{P}{\tau} = \Phi(V)$$ (29)

In other words, the *P*-scaling law of the VCV18-1 system is fully defined by the balance between positive
and negative feedbacks. For $\tau = 50$ kyr, $V = 0.63$, $\Phi = 2$. The corresponding 100-kyr-period auto-
oscillations of the system (22) – (23) and its positive-versus-negative feedback loop are shown in Fig. 2
(e, f).
**2.4 VDP model**
We now consider the VDP model, which is a variant of the historical van der Pol model (1922) used
by De Saedeleer et al. (2013) and Crucifix (2013) to study synchronization properties of ice ages.
$$\frac{dx}{dt} = \frac{-\beta - y}{\tau}$$ (30)
$$\frac{dy}{dt} = \frac{\alpha}{\tau}\left(y - \frac{y^3}{3} + x\right)$$ (31)



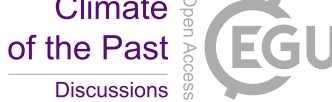

Here all variables and parameters, except time and timescale $\tau$, are adimensional. Variable $x$ is a proxy for
the glaciation, and variable $y$ represents the rest of the climate. Since $\frac{\tau}{\alpha} \ll \tau$, for longer, glaciation-like
processes, we can re-write equations (30) – (31) as:
$$\frac{dx}{dt} = \frac{1}{\tau}\left(-\beta + x - \frac{y^3}{3}\right) \qquad (32)$$
$$y - \frac{y^3}{3} + x = 0 \qquad (33)$$
The equation (33) defines the "critical manifold" (Guckenheimer et al, 2003). The system (32) – (33)
describes VDP "slow" dynamics between two glacial-interglacial bifurcation points. To get VDP time
series, we solve the non-idealized system (30) – (31), and we use the system (32) – (33) to visualize
system's positive, $x$, and negative, $\frac{y^3}{3}$ feedbacks. Schematically, the dynamical system (30) – (31) is
shown in Fig. 1 (d). The period of system (30) – (31) variability is the function of two governing
parameters:
$$P = \psi(\tau, \beta) \qquad (34)$$
Only parameter $\tau$ is dimensional, and therefore according to $\pi$-theorem:
$$\frac{P}{\tau} = \Psi(\beta) \qquad (35)$$
The amplitudes of VDP variables are defined by the critical manifold (33) that does not contain
parameters $\tau, \beta$ and therefore both $x$- and $y$-amplitudes do not depend on model parameters. Consequently
the amplitudes of positive and negative feedbacks do not depend on them either. In fact, the $x$-amplitude
in VDP model is always ~0.8 and $y$-amplitude is always 2. Therefore the ratio of the amplitude of the
positive feedback, $x$, to the amplitude of the negative feedback, $\frac{y^3}{3}$, is always $V = 0.3$. In summary:
$$V = const \qquad (36)$$
The property (36) makes VDP model to be fundamentally different from all other models in this
study. In all other models, the $V$-number is the function of model's governing parameters and, under
different scenarios, it changes when parameters change. In the VDP model, the $V$-number is pre-defined
by the model's structure. Consequently, the scaling law (35) does not contain any $V$-number and thus it
cannot match the scaling law (29). Therefore, *there is no physical similarity* between the VCV18-1 and
the VDP models.
The auto-oscillations of the system (30) – (31) and positive-versus-negative feedback evolution are
shown in Fig. 2 (g, h). For $\tau = 50$ kyr and $\beta = 0.3$, $\Psi = 2.2$. Fig. 2 (e) and Fig. 2 (g) show well why the
phenomenological approach may be misleading. For the same internal timescale of 50 kyr, VCV18-1 and
VDP models both produce asymmetrical (slow growth and fast retreat) glaciation time series with the
respective periods $P$ close to 100 kyr, but this occurs because of very different physics: in the VDP model
the positive feedbacks are much weaker (as we already know, $V = 0.3$, always) than in the VCV18-1
model ($V = 0.63$). Most importantly, this discrepancy cannot be changed, because the VDP model is
rigid in this regard.
**2.5 LP22 model**
The LP22 model is described by two differential equations, first, for the growing ice volume,



$$\frac{dv}{dt} = -\frac{I}{\tau_i} + \frac{1}{\tau_g} \qquad (37)$$
and another one for the diminishing ice volume
$$\frac{dv}{dt} = -\frac{I}{\tau_i} - \frac{v}{\tau_d} \qquad (38)$$
Here, $v$ and $I$ are normalized ice volume and astronomical forcing, correspondingly; $\tau_i$, $\tau_g$, and $\tau_d$ are
dimensional timescales. Additionally, if $I < I_0$ the system switches from equation (38) to equation (37),
and if $I + v > V_0$, the system switches from equation (37) to equation (38). Though the original LP22
model does not consider its evolution without astronomical forcing, oscillations still occur when $I = 0$ and
the equation-switching conditions are, correspondingly, $v \leq V_1$ ($V_1$ is the minimal, interglacial, volume)
and $v \geq V_0$. Schematically, the dynamical system (37) – (38) with $I = 0$ is shown in Fig. 1 (e). The period
of auto-oscillations is a function of four parameters:
$$P = \psi(V_0, \tau_g, V_1, \tau_d) \qquad (39)$$
The parameter $V_1 \ll V_0$ can be settled as a constant, therefore:
$$P = \psi(V_0, \tau_g, \tau_d) \qquad (40)$$
If we select $\tau_g$ as an independent-dimension parameter, then according to $\pi$-theorem:
$$\frac{P}{\tau_g} = \Psi(V_0, \tau_d/\tau_g) \qquad (41)$$
The equation (37) describes a linear ice volume growth implying zero net feedback. This doesn't indicate
the absence of feedbacks. Indeed, if we are ready to accept that LP22 model is more than just a successful
fit to empirical data, then for the growing ice sheet, the equation (37) should be consistent with the
dimensional total mass balance
$$\frac{dv}{dt} = \hat{a}S \qquad (42)$$
i.e., changes of ice volume are equal to mass influx $\hat{a}$ *accumulated over its area $S$*. If we multiply and
divide $\hat{a}S$ by ice thickness $H$, the total mass balance (42) becomes
$$\frac{dv}{dt} = \frac{v}{\tau_g} \qquad (43)$$
where $\tau_g = H/\hat{a}$. The equation (43) tells us that the positive feedback $\frac{v}{\tau_g}$ must be present in the growing
ice sheet: it relates to the area growth with thickness. Its absence in equation (37), therefore suggests that
$\hat{a}$ has another component that completely compensates for $\frac{v}{\tau_g}$, and yet another component that is inversely
proportional to $S$ (e.g., continentality effect). Hence, the system (37) – (38) must be written as follows to
have physical meaning:
$$\frac{dv}{dt} = \frac{1}{\tau_g} + \frac{v}{\tau_g} - \frac{v}{\tau_g} \qquad (44)$$
$$\frac{dv}{dt} = -\frac{v}{\tau_d} \qquad (45)$$





Schematically, the dynamical system (44) – (45) is shown in Fig. 1 (f). The amplitude of the positive
feedback is $\frac{V_0}{\tau_g}$ and the amplitude of the negative feedback is $max\left\{\frac{V_0}{\tau_g};\frac{V_0}{\tau_d}\right\}$. Since $\tau_g > \tau_d$, the *V*-number
therefore is equal to:

$V = \frac{\tau_d}{\tau_g}$ (46)
Accordingly, the *P*-scaling law (41) for LP22 model can be written as
$\frac{P}{\tau_g} = \Psi(V_0, V)$ (47)
For example, for $\tau_g = 50$ kyr, $V_0 = 1.5, V_1 = 0.2, \tau_d = 20$ kyr, we get $V = 0.4$ and $\Psi = 2$. As we
have established before, for VCV18-1 model, for $\tau = 50$ kyr, $V = 0.63$, $\Phi = 2$. Therefore, we can talk
about *physical similarity* between VCV18-1 and LP22 models in terms of the *V*-number.
The similarity between VCV18-1 and LP22 models becomes very visual in Fig. 2(f) describing the
positive-versus-negative feedback loops. It can be observed that in VCV18-1 model, during much of ice
growth, positive and negative feedbacks also completely compensate each other. In fact, we can consider
LP22 model as being an approximation of the VCV18-1 feedback loop by a triangle. Indeed, the LP22
positive and negative feedbacks compensate each other during ice advance and when the critical value of
$v$ (i.e., $V_0$) is achieved, the system instantly migrates to the single dominant negative feedback. Simply
speaking, these two models are as similar as the shape of the LP22 feedback-loop triangle in Fig. 2(f) is
similar to the shape of the VCV18-1 feedback loop, since the *V*-number is the ratio of its horizontal
dimension to its vertical dimension.

**3. Conclusions**

Nikolai Gogol would have said: "A magic apple tree may grow golden apples…but not pears".
Phenomenological models may not be bounded by a specific physics but they have to be consistent with
the laws of physics. The concept of physical similarity is what allows us to be vigilant about such
consistency. Accordingly, we started our presentation with the question: Are phenomenological
dynamical paleoclimate models physically similar to Nature? We demonstrated that, though they may be
remarkably accurate in reproducing empirical time series, this is not sufficient to claim physical similarity
with Nature until similarity parameters are considered. We illustrated such a process of diagnostics of
physical similarity by comparing three phenomenological dynamical paleoclimate models with the more
explicit model that played the role of parent dynamical system. Though the nomination of the VCV18
model to serve as a proxy of the parent dynamical system can, indeed, be questioned, and the
developments of better proxies should be encouraged, we nevertheless believe that the diagnostics of
physical similarity, we have described, should become a standard procedure before a phenomenological
model can be utilized for interpretations of historical records or for future predictions. In other words,
claiming a model to be a phenomenological one is not an indulgence but a liability.
The results of the analysis are summarized in Fig. 3. Here, the physical similarity is visualized as
proximity between the models in the $\left(\frac{\varepsilon}{\hat{a}}, V\right)$ space. The positive-versus-negative feedback diagrams
provide additional insight. It can be observed that the original VCV18 and G23 models are very different
from VCV18-1, VDP, and LP22 models not just because the latter may generate 100-kyr cycle without
orbital forcing. The mechanism of ice disintegration is very different in these two groups of models. In
VCV18-1, VDP, and LP22 models, the disintegration of ice sheets happens when the negative feedback
suddenly becomes dominant and destroys an ice sheet. In both VCV18 and G23 models, the
disintegration is due to the additional, fast, positive feedback, which is small during most of the ice-



growth period, but eventually becomes strong enough to boost the orbital forcing that attempts ice
destruction.

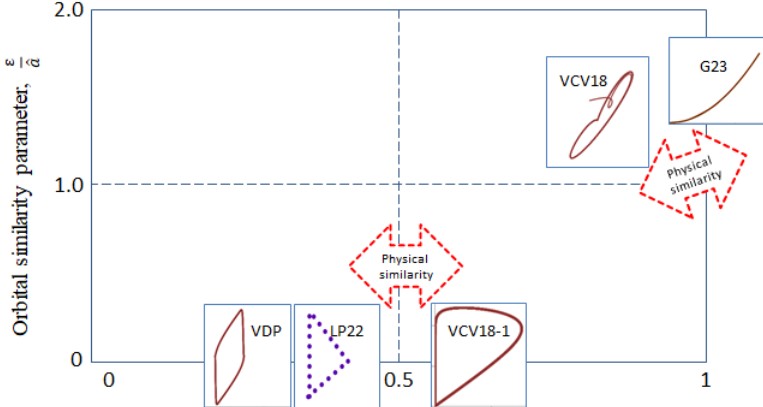


**Figure 3.** Physical similarity diagnostics in the $\frac{\varepsilon}{\hat{a}} - V$ space for the obliquity-period doubling of the
VCV18 and G23 models and for 100-kyr auto-oscillations of the VDP, LP22, and VCV18-1 models
LP22 and G23 models can be considered to be physically similar to particular versions of the VCV18
model, which amounts to saying that they may be derived from basic laws of physics under some
reasonable physical assumptions. These findings boost the physical viability of these phenomenological
models, but this is not unconditional and there are clear boundaries of these phenomenological models
physical legitimacy. For LP22, the ratio of feedback's amplitudes, the $V$-number, is also the ratio of
timescales. For the Late Pleistocene and for the Early Pleistocene, all timescales are likely to be different
and physical similarity therefore would need to be re-examined for each period separately. For G23
model, physical similarity was found only for the obliquity-range forcing.
Generally speaking, our observation that the VDP model is not physically similar to the simplified
version of the VCV18 model is not a final verdict. It is indeed an indication that VDP is not based on *ice
physics*, but there may be other physical phenomena that may provide physical legitimacy to VDP model.
We are a bit skeptical though that such phenomenon can easily be found, because it would need to be
constrained in the same way as VDP is. Specifically, its ratio of positive and negative feedbacks must to
be fixed to a specific value that never changes.
As a final conclusion, we agree with Saltzman's (2002) proposal that "the essential slow physics is to
be sought in the low-order models." We observe though that "essential slow physics" that can be derived
from phenomenological models is limited to orbital and terrestrial timescales, to ratios of amplitudes of
orbital and terrestrial forcings, and to ratios of amplitudes of positive and negative feedbacks. This is as
much as phenomenological models can offer, and therefore, we deviate from Saltzman's (2002) further
idea that more explicit models should be tuned to satisfy a best phenomenological model. Instead, we
propose to use available physical models for diagnostics of physical-similarity hypothesis that needs to be
either confirmed or rejected.
Of course, encoding empirical data in a simple mathematical statement will always remain a tempting
possibility. As Grigory Barenblatt (2003) said "Applied mathematics is the *art* of constructing
mathematical models of phenomena in nature…" This means that there are no strict rules on how a piece
of mathematical "art" needs to be produced. Therefore, we do not attempt here to discourage our fellow
"artists" from alluding to phenomenological models. Our goal instead was to remind them about
phenomenological models' limitations and to suggest how these limitations may be addressed.




**Author contributions:** MYV conceived the research (Verbitsky, 2022b), developed the formalism, and
wrote the first draft of the manuscript. The authors jointly discussed the findings and contributed equally
to the editing of the manuscript.
**Competing interests:** The authors declare that they have no conflict of interest.
**Acknowledgement:** We are grateful to Andrey Ganopolski for discussions and generous insight into G23
model that includes time series of Fig. 2(c, d), and to Dmitry Volobuev for his help in digitizing VCV18.

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
