# Peer review of "Do phenomenological dynamical paleoclimate models have physical similarity with "Nature"? Seemingly, not all of them do."

_Climate of the Past, 2023_

## Referee Comment (RC2)

The authors aimed to measure the physical similarity between several phenomenological dynamical paleoclimate models and the real world. They proposed that a model should have a link with known physical assumptions, rather than a statistical description of proxy data. Overall, the manuscript is interesting and may have important implications for the future development/evaluation of phenomenological models. As I am not an expert on this issue, I fully agree with the comments from Reviewer#1. However, I still have some comments that may be considered to improve the quality of the manuscript.

1. Figure 2 shows a specific case of the models used here. I am wondering how you choose the values of each parameter (e.g., why obliquity-period doubling is used?). Please clarify.

2. The physical similarity is measured in the $\varepsilon/\hat{a}$ -V space. I am just wondering what are the limitations of the proposed method? For example, G23 model clearly differs from the VDP model in the $\varepsilon/\hat{a}$ -V space. Are there possibilities that G23 model shares similarities with VDP model in some aspects?

3. The positive-vs-negative feedback loops are important to understand the periodicity of glacial-interglacial cycles. Could you add additional discussions on the differences in the loops between each model?

---

## Author Response (AR1)

Dear Prof. Zhang,

We are sincerely grateful to our two anonymous reviewers. We adopted all their comments and hope that the manuscript is now close to their expectations. All changes are marked red.

Respectfully,

Mikhail Verbitsky and Michel Crucifix

**Response to Anonymous Referee #1**

Dear Anonymous Referee #1,

We are grateful for your insightful review that will help us to improve the manuscript. Your *verbatim* comments are below (in bold), each followed by our response.

**General comments: This manuscript has the potential to provide an important contribution to Ice Age modelling.**

**Response:** We are encouraged by your recognition of such important potential.

**General comments (cont'd): Building on their previous work (Verbitsky & Crucifix, 2020; 2022), the authors investigate the physical similarity of different models using dimensional analysis. A key finding is that although all these models are structurally dissimilar, they share a dependence of the modeled periodicity on the dimensionless "V" parameter or the ratio between positive and negative feedbacks in the system.**

**Response:** Your observation is correct: Indeed, finding the key similarity parameter that allowed us to quantitatively compare phenomenological models with the model that we nominated to serve as the proxy of the parent dynamical system is very important. Another key finding, that you have not mentioned, is that not all phenomenological models have actually "passed" the similarity test and therefore did not demonstrate a link with known physical assumptions. This finding has in our view philosophical-level consequences: successful replication of empirical time series is not sufficient to claim physical similarity with Nature.

**Having said all this, I do think that the authors will need to address the issues discussed below before the manuscript will be ready for publication.**

**Specific comments: I have two major concerns that the authors will need to address in a compelling manner:**

**1) The authors make it seem as if they are comparing models to Nature when they are really comparing different models to each other. And yes, even the Verbitsky et al. (2018) model is not the same thing as Nature. The authors should be transparent about this, which starts with the title. I suggest changing it from "Do phenomenological dynamical paleoclimate models have physical similarity with Nature? Seeming, but not all of them" to something like "Structural similarities and differences between paleoclimate models of glacial-interglacial dynamics". Then, the Abstract and Introduction could be framed around questions such as "To which extent are different paleoclimate**

models of glacial-interglacial dynamics physically similar?" and "Are there any shared dimensionless quantities playing key roles in all these models? If so, then finding values for these quantities should be a central objective of future research into glacial-interglacial dynamics."

**Response:** Your concern is fair and it is well understood. In fact, we tried hard to be very transparent in that regard: (a) We introduced the notion of the parent dynamical system (lines 63-69), (b) then we formulated requirements to it and introduced the VCV18 model as a candidate, a proxy, for the parent system (lines 108-113, 132-143), (c) we even suggested that "the nomination of the VCV18 model to serve as a proxy of the parent dynamical system can, indeed, be questioned, and the developments of better proxies should be encouraged" (lines 427-429). We agree with you though that there is still room for improvement, therefore…

**Action:** Your questions "To which extent are different paleoclimate models of glacial-interglacial dynamics physically similar?" and "Are there any shared dimensionless quantities playing key roles in all these models? If so, then finding values for these quantities should be a central objective of future research into glacial-interglacial dynamics" are well formulated and we will definitely discuss them in the introduction and conclusions sections of the revised paper. We are a bit reluctant though to accept your new title suggestion because, in our opinion, it shifts the focus of the paper. We believe that it is not so much about how similar models are, but about how similar are models, that are merely a statistical description of the data (phenomenological models), and a model that was derived from the basic laws of physics. We will make sure though that the Nature in the title will become quote-unquote "Nature".

**Done**: New lines 2, 23-28, 508-510

**2) The authors use the Buckingham pi theorem to answer the identify the dimensionless parameters affecting the period of the system in each of the models. All well and good, but what do we really learn from this about glacial-interglacial dynamics? I think much more insight could be gained if the authors would identify the actual relationships between the parameters and the period of the model systems (i.e., the psi and phi functions). It should not be too difficult to find decent approximations of these relationships through simulations, given that the models under consideration are rather simple and computationally cheap to run. Such an exercise would also give much deeper insight about the physical similarity between the different models. For example, the period may depend on "V" in two models, but the scaling may be V^2 in one model and V^3 in the other. Then, one can ask where these different scalings originate from and whether any of these scalings could be tested against observational data.**

**Response:** Your observation is correct – we do not provide the most explicit form of the scaling laws, specifically, we do not convert

$$\Pi = \Phi(\Pi_1, \Pi_2, \dots, \Pi_i, \dots, \Pi_m) \tag{AC1}$$

into

$$\Pi = \Pi_1^{\alpha_1} \Pi_1^{\alpha_2} \dots \Pi_1^{\alpha_m} \tag{AC2}$$

Also, we do not convert

$$\pi = \Psi(\pi_1, \pi_2, \dots, \pi_i, \dots, \pi_q) \tag{AC3}$$

into

$$\pi = \pi_1{}^{\beta_1} \pi_1{}^{\beta_2} \dots \pi_1{}^{\beta_q} \qquad \text{(AC4)}$$

We limited our scope to the demonstration that in some models

$$\Psi \approx \Phi \qquad \text{(AC5)}$$

when

$$\pi_i \approx \Pi_i, \qquad \text{(AC6)}$$

i.e., $\pi_i$-physics in the model is as significant as the $\Pi_i$-physics of "Nature". It allowed us to "delineate a model that is merely a statistical description of the data, from a model that can be claimed to have a link with known physical assumptions" (lines 20-22). To our knowledge, *this result is novel.*

Importantly, *the condition (AC6) can also be considered as a constraint on the parameterization structure of a model, and as such, it is indeed also a new insight.*

Though we agree with you that the finding of explicit scaling laws (AC2) and (AC4) is an important exercise, we believe that this should be a separate study because it may be a bit more challenging than it seems to be. For example, gradual increase of the *V*-number in the VCV18 model includes a bifurcation from the obliquity period to the obliquity-period doubling (Verbitsky and Crucifix, 2020). Moreover, the *V*-number in the VCV18 model is a conglomerate similarity parameter, composed by 5 similarity parameters and, though the bifurcation is imminent, the critical value of the *V*-number that starts a bifurcation depends on which specific similarity parameters produce the *V*-number change. For example, the bifurcation that is caused by increased *V*-number due to the intensification of the positive feedback and caused by changes of the governing parameter responsible for the positive feedback may start at the critical *V*-number value that is different from the critical *V*-number value caused by the weakening of the negative feedback (Verbitsky, 2022). Therefore, to get a complete picture, the number of experiments needed for spanning the full parameter space can be overwhelming.

However, to address your concern, a few steps in this direction can be done, indeed, without much of computer power. First, we have run a few additional experiments with the VCV18-1 model:

$$\frac{dH}{dt} = \hat{a} + \kappa H^4 - c\theta \qquad \text{(22)}$$

$$\frac{d\theta}{dt} = \frac{H^4 - \theta}{H/\hat{a}} \qquad \text{(23)}$$

Specifically, we numerically measured the dimensionless period of auto-oscillations (*P*-scaling law) $\frac{P}{\tau} = \Phi(V)$, changing parameter *c*, and thus gradually changing the balance between positive and negative feedbacks. The results of these additional experiments are presented in Figure AC1 together with the *P*-scaling law for LP22 model $\frac{P}{\tau_g} = \Psi(V_0, V)$ that can be estimated in a very straightforward manner:

$$P = V_0 \tau_g + \tau_d, \text{ or}$$

$$\frac{P}{\tau_g} = V_0 + \tau_d/\tau_g \text{ , or}$$

$$\frac{P}{\tau_g} = V_0 + V \qquad\qquad\qquad\qquad\qquad\qquad\qquad\qquad\qquad\text{(AC7)}$$

And finally, as an illustration, we also made a few experiments with the VCV18 model, gradually changing the *V*-number and measuring the corresponding *P*-scaling law. For this specific illustration, the *V*-number was modified by changing the strength of the positive feedback through the coefficient $\gamma$ in the equation (10). The results are also presented in Figure AC1.

[Figure]

**Fig. AC1.** The *P*-scaling laws of VCV18-1 (blue, the dotted line is its trendline) and of LP22 (brown) models. The green dot marks the scaling law for the VDP model. The red line illustrates a bifurcation from the obliquity period to the obliquity-period doubling in the VCV18 model.

Interestingly, though the VCV18-1 scaling law $\frac{P}{\tau} = \Phi(V)$ and LP22 scaling law $\frac{P}{\tau_g} = \Psi(V_0, V)$ were produced absolutely independently, the LP22 scaling law coincides almost perfectly with the VCV18-1 scaling-law trendline. This closeness of the VCV18-1 and LP22 scaling laws supports our previous assertion that we can consider LP22 model as an approximation of the VCV18-1 (lines $407 - 415$). The new figure also shows that better articulated positive feedbacks (i.e., increased *V*-number) lead to longer periods of both auto-oscillations and periods of the system response to the orbital forcings.

**Action:** We will include parts of the above conversation in the paper. In doing so, we will rely on our editor advice to reasonably constrain the divergence from the original scope.

**Done:** New Fig. 3 and new lines 428-434

**Technical corrections**

**l. 16:** "...similar with the..." -> "...similar to the..."

**l. 55:** "Dynamical system's theory tell us why..." -> "Dynamical Systems Theory tells us why..."

**l. 59:** "...phemenon..." -> "...phenomenon..."

**l 79:** "...parameters, let say Pi_1, is..." -> "....parameters, say Pi_1, is..."

**l. 375:** "...according to pi-theorem:" -> "...according to the pi-theorem:"

**Action:** Thank you! All corrections will be taken care of.

**Done:** New lines 16, 62, 66, 86.

**Response to Anonymous Referee #2**

Dear Anonymous Referee #2,

We are grateful for your insightful review that will help us to improve the manuscript. Your *verbatim* comments are below (in bold), each followed by our response.

**General comments: The authors aimed to measure the physical similarity between several phenomenological dynamical paleoclimate models and the real world. They proposed that a model should have a link with known physical assumptions, rather than a statistical description of proxy data. Overall, the manuscript is interesting and may have important implications for the future development/evaluation of phenomenological models.**

**Response:** We are grateful for your vision that "the manuscript is interesting and may have important implications for the future development/evaluation of phenomenological models."

**Specific comments: As I am not an expert on this issue, I fully agree with the comments from Reviewer#1. However, I still have some comments that may be considered to improve the quality of the manuscript.**

**1. Figure 2 shows a specific case of the models used here. I am wondering how you choose the values of each parameter (e.g., why obliquity-period doubling is used?). Please clarify.**

**Response:** We *do not prescribe* the obliquity-period doubling. Instead we calculate VCV18 and G23 models responses (with the reference values of parameters published in VCV18 and G23) to the external sinusoidal forcing of the obliquity period and find that in both cases the period of response is twice the forcing period. Most importantly, it happens with similarity parameters, both the ratio of astronomical forcing amplitude to the terrestrial growth rate and the positive-to-negative feedback ratio, having reasonably close values in both models. Therefore we conclude that there is physical similarity between VCV18 and G23 models in terms of the above similarity parameters.

**Action:** We will further clarify this in the text.

**Done:** New lines 490-492

**2. The physical similarity is measured in the $\varepsilon/\hat{a}$ -V space. I am just wondering what are the limitations of the proposed method? For example, G23 model clearly differs from the VDP model in the $\varepsilon/\hat{a}$ -V space. Are there possibilities that G23 model shares similarities with VDP model in some aspects?**

**Response:** Fig. 3 is just an attempt to present most important results visually in one diagram. Since it is 2-dimensional, it may demonstrate similarity (or its absence) in terms of only 2 similarity parameters. It was sufficient for us, since we found similarity between phenomenological and physical models in terms of two similarity parameters, the ratio of astronomical forcing amplitude to the terrestrial growth rate and the positive-to-negative feedback ratio. This space wouldn't be sufficient if more similarity parameters needed to be visually presented. For example, equation (21) shows that G23 model depends on the similarity parameter that is the ratio of the orbital period $T$ to the internal timescale $1/a$. This similarity parameter is absent in VCV18 model. This absence of similarity is not apparent from Fig. 3, but it is discussed in lines 240-254.

**Action:** We will further clarify this in the text.

**Done:** Updated Fig. 4 and new lines 493-500

**3. The positive-vs-negative feedback loops are important to understand the periodicity of glacial-interglacial cycles. Could you add additional discussions on the differences in the loops between each model?**
**Response:** We agree with your observation. Indeed, "The positive-vs-negative feedback loops are important to understand the periodicity of glacial-interglacial cycles". While, these loops is the central part of VCV18-1 and LP22 models comparison (lines 407-415), in other cases such discussions, we admit, may be too brief.

**Action:** We will add additional discussions of the feedback-loops diagrams.

**Done:** New lines 267-270, 478-489

[revised manuscript text omitted]